# Neonicotinoids: Spreading, Translocation and Aquatic Toxicity

**DOI:** 10.3390/ijerph17062006

**Published:** 2020-03-18

**Authors:** Mária Mörtl, Ágnes Vehovszky, Szandra Klátyik, Eszter Takács, János Győri, András Székács

**Affiliations:** 1Agro-Environmental Research Institute, National Agricultural Research and Innovation Centre, H-1022 Budapest, Herman O. u. 15, Hungary; klatyik.szandra@akk.naik.hu (S.K.); takacs.eszter@akk.naik.hu (E.T.); szekacs.andras@akk.naik.hu (A.S.); 2Department of Experimental Zoology, Centre for Ecological Research, Balaton Limnological Institute, H-8237 Tihany POB 35, Hungary; gyori.janos@okologia.mta.hu

**Keywords:** acetycholine esterase (AChE), *Daphnia magna*, dosages, glutathione-S-transferase (GST), guttation, mollusk, nAChR, neonicotinoids, neurotoxicity, water pollutant

## Abstract

Various environmental and ecotoxicological aspects related to applications of neonicotinoid insecticides are assessed. Dosages of neonicotinoids applied in seed coating materials were determined and are compared to other applications (spray and granule). Environmental levels in soils and affecting factors in translocation are discussed. Excretion of neonicotinoids via guttation from coated maize seeds up to two months upon emergence, as well as cross-contamination of plants emerged from non-coated seeds or weeds nearby have been demonstrated. Contamination of surface waters is discussed in scope of a worldwide review and the environmental fate of the neonicotinoid active ingredients and the formulating surfactant appeared to be mutually affected by each other. Toxicity of neonicotinoid active ingredients and formulations on *Daphnia magna* completed with some investigations of activity of the detoxifying glutathione S-transferase enzyme demonstrated the modified toxicity due to the formulating agents. Electrophysiological results on identified central neurons of the terrestrial snail *Helix pomatia* showed acetylcholine antagonist (inhibitory) effects of neonicotinoid insecticide products, but no agonist (ACh-like) effects were recorded. These data also suggested different molecular targets (nicotinergic acetylcholine receptors and acetylcholine esterase enzyme) of neonicotinoids in the snail central nervous system.

## 1. Introduction

The first neonicotinoid insecticide, imidacloprid (IMI) was introduced in 1991 as a result of Japanese and European research activities, followed by nitenpyram and acetamiprid (ACE) in 1995, thiamethoxam (TMX) in 1997, thiacloprid (TCL) in 1999 and clothianidin (CLO) in 2002 [1]. Other active ingredients (AIs) were also commercialized and a research program for the discovery of new neonicotinoid type insecticides was established also in China [2]. Their large-scale deployment in seed treatments [3] as preventive pest management in field crops (e.g., maize, sunflower, cotton) has driven their rapid increase in worldwide use. Due to their broad-spectrum activity on a range of insects, the introduction of neonicotinoids was initially regarded as a substantial improvement in insect control, and their use was considered to exert only a minimal impact on wildlife and the environment. Their application in seed coatings represented a reduced risk compared to spray applications, and their outstanding insect-selective toxicological profile was also advantageous compared to AIs used earlier (e.g., organophosphates). After the coated seeds germinate, neonicotinoid molecules are rapidly taken up by the roots and transported into the young shoots and leaves. The systemic movement, along with a long residual activity in the plant ensured effective protection against the early season insect pests during the vulnerable, young growth stages of plant development.

Nonetheless, their effective insect specificity soon manifested itself also on non-target species [4], including pollinators, particularly honeybees [5]. Due to their extended use and mobility, residues of neonicotinoids became ubiquitous contaminants in surface water [6], and appeared in honey [7] and other agricultural products [8]. Their dissipation time from soil is variable [9], but long dissipation times (DT_50_ = 143–1001 days) were reported e.g., for CLO. The wide-scale use of neonicotinoid insecticides, especially as seed treatments, have raised concerns regarding environmental load, as well an impact on pollinators, biodiversity and ecosystems. Their negative effects reported on non-target insect species led gradually to the restriction of the use of three of the most important neonicotinoids, CLO, IMI and TMX in the European Union (EU) in 2013 [4]. Upon collection and peer review of scientific information [5] including environmental risk assessment, particularly for bees, application of these AIs was banned under field conditions in the EU [6,7,8], and their use remained authorized only in closed greenhouses. Other neonicotinoid AIs are still in use in Europe. ACE remained authorized and applied in pest control in various forms of administration, and TCL remained authorized as the only neonicotinoid for seed coating of maize in the EU. The US EPA cancelled the registration of 12 neonicotinoid plant protection products (PPPs) in 2019, scheduled the detailed review of neonicotinoids for 2020, suggests restrictions on applying these insecticides to blooming crops, and advises against their household use on residential lawns and turf, for which application proposes a ban on IMI [10].

Due to the strict restrictions on CLO, IMI and TMX, replacement agrotechnological solutions became a pressing need in the EU, and are also considered in the US. Possible alternative pest management strategies were recently presented [11], to eliminate neonicotinoid-based chemical pest control in cropping systems. These tools provide considerable reduction in the utilization rate of these insecticides by replacing their prophylactic use with applications justified by on-site risk assessment, on the one hand, and by a financial compensation system of unpredicted damages by crop insurance, on the other hand. Similar measures have been proposed for maize in the US, where 71% to nearly 100% of the maize cultivation fields are exposed to neonicotinoids, while only a small fraction of that area would likely be exposed to applied insecticides if seed coatings with neonicotinoids were eliminated [12]. For some pests, data are available to predict years in which heavy infestations may be more common (e.g., pests with multi-year life cycles), although forecast damages can be difficult in given cases due to poor historical experience on temporal pest population dynamics [13]. Insecticides are rarely needed to control early-season pests (e.g., in maize), and related crop loss can be largely eliminated by non-chemical and agroecological methods [11], including mating disruption, attract-and-kill strategies using biological tools (e.g., microbial agents or biological control with flowers grown on the bunds of rice fields), natural or food-derived insecticides, insect repellents (e.g., nettle extract) and trap attractants. Farmers generally relied on alternative seed treatments or more soil/foliar treatments in the first growing season after the restrictions took effect [14]. In France, the most common alternative to neonicotinoids has been the use of other chemical insecticides (mostly pyrethroids) [15], despite the fact that at least one non-chemical alternative method (e.g., microorganisms, semiochemicals or surface coating) was available in most of the cases.

The main objective of this work is to present environmental features of neonicotinoids including their surface water contamination potential, persistence in soil and plants, their translocation to the guttation liquid, as well as toxicity to non-target aquatic and terrestrial organisms (the water flea *Daphnia magna* and snail *Helix pomatia*). There is a special focus in this study on the use of these insecticides as seed coating materials including the corresponding dosages, because seed coating became the predominant agrotechnological application mode and at the same time the main source of contamination.

## 2. Use of Neonicotinoids as Coating Materials, Dosages

Before the approval of neonicotinoids in seed treatment, numerous insect pests were controlled with the use of in-furrow granular insecticides. Seed coating is regarded as a precise delivery method of pesticides, which is therefore more effective compared to other plant protection product (PPP) applications (e.g., spraying). Neonicotinoids persist long enough to provide an effective control on a broad spectrum of insects, which led to their increased adoption by growers. As for maize, an estimated annual average of 45% and 25% of the total crop cultivation in the US was treated with CLO and TMX, respectively (with annual maxima of 65% for CLO and 45% for TMX in any given year) [16], and corresponding average proportion of the use of seeds coated with CLO or TMX are 46%, 33%, 24% and 18% for sugar beet, cotton, sorghum and soybean, respectively [16]. In accordance with these maximum rates, nearly 100% of maize in the US and canola in Canada is cultivated from seeds coated with neonicotinoids [17] and often fungicides as well.

One of the first signs of the environmental problem caused by neonicotinoids was detected in Germany in the spring 2008, when abrasion of active substance from treated seeds during sowing of maize led to mass poisonings incidents of bees [18]. Thereafter, substantial efforts were made to minimize the environmental impacts of these crop protection products. Improvements in seed coating technology (e.g., application of polymers) and modification of pneumatic planters have been introduced to reduce abrasion of the seed coating material during planting. Polymers in seed treatment not only improved coverage of the seed surface with the AI, but also reduced dust-off from treated seeds and improved the drop of seeds by the planter.

An assessment of the application rates of commercially coated maize seeds (for experimental details see ref [19]) and on-farm (home) coated seeds revealed interesting findings. For maize, the recommended treatment rates range between 0.25 and 1.25 mg AI/kernel [20]. The measured data were compared with the recommended dosages (see Table 1), and insecticide AI content for commercial seeds were generally in accordance of the recommended doses, however doses were lower for earlier commercial seeds (CS series: CS-1 to CS-3) and intentionally lower for on-farm coated seeds (HC series: HC-1 to HC-3). Lower dosages (0.25 mg/seed) were probably applied earlier, but due to the resistance problems, maintaining efficacy gradually required higher dosages, as generally observed for any widely applied PPPs. Higher rates are required also to provide protection against the corn rootworm and billbug, compared to other insect species (e.g., wireworm). Relative standard deviations (RSDs) were typically around 20% for seeds purchased in 2007–2009. Somewhat lower values (around 10%) were observed for more recent seeds (2013–2015), but in a single case (CS-4) a substantially higher (54.3%) standard deviation was determined. Comparing the results measured for an earlier seed (CS-3) and for CS-4 (see Figure 1) both coated with CLO, the ratio between the highest and lowest values were 4.73 and 2.16 for CS-4 and CS-3, respectively. A total of 6 of the 15 seeds were out of the RSD for CS-3, but 7 of 15 seeds were out or near to RSD for CS-4 and 3 of the 7 values were significantly higher than the range based on RSD. Ageing of the coating material as well as the improvements of seed coating technology could play a role in this phenomenon. High differences between application rates (maximum/minimum = 4.73) might influence the local concentrations in soil, thus, the ingredient uptake.

Commercial seed treatment facilities utilize the latest technologies to assure the accuracy of the application of seed treatment products compared to the on-farm seed treatment practice, where less efficient equipment is likely to be used, potentially resulting in higher exposures. Dosages in on-farm seed coatings were considered intentionally lower, but RSD values, in contrast to our expectations, were not significantly higher, than those for commercial seeds (It has to be noted, that on-farm seed treatment is specifically prohibited in the US e.g., for CLO [20] (but not for TMX), and the use of less efficient equipment is likely to result in higher occupational exposures).

Comparing the recommended doses for granules and spray applications with those of for seed coating, it is concluded that the doses are very similar in all cases and environmental loads are practically the same [19]. Dosages of neonicotinoids in seed coating correspond to 30–85 g AI/ha (0.6–1.22 mg AI/seed at 50–70 thousand (maize) plants/ha). A similar calculation by the US EPA [20], based on a maximum planting rate of 86,500 maize seeds (kernels)/ha and treatment rates of 0.25 or 1.25 mg AI/kernel, resulted in application rates between 21.7 and 108.7 g AI/ha for CLO. Typical dosages in spray and in soil granule applications are 20–70 g AI/ha (20–70 mg AI/L at 1000 L/ha) and 110 g AI/ha (10 g AI/kg at 11 kg/ha), respectively [19]. Seed coating is more favorable in terms of pesticide consumption only if spray applications are needed to be used several times during the vegetation period (the number of registered applications is limited to two sprayings per season). The only difference is that granules and coating materials are in a solid form, therefore, ingredients are present at higher concentrations in soils locally compared to spraying agents. Seed treatment considerably reduces soil surface exposure as well, compared to in-furrow and surface applications, but contamination cases were detected for the use of seed coating as well [21].

## 3. The Presence of Neonicotinoids in Soil

High amounts of neonicotinoids are applied to crops worldwide, yet a large proportion of the applied substances remains in the soil and potentially contaminates all environment compartments (air and dust, soil, water, sediments and plants). Reported levels in the soils were typically in the low ng/g range with maximum values up to 30 ng/g, but from soils of cocoa plantation in Ghana up to 251 ng/g IMI was measured [22]. For example, neonicotinoid residues were measured at an average concentration of 4.0 ng/g (ranging between 0.07 and 20.3 ng/g) for soil samples taken before planting of coated maize seeds in Southwestern Ontario [21]. Average levels increased to 9.9 ng/g (ranging between 0.53 and 39.0 ng/g) for samples taken immediately after planting from the top 5 cm of the soil. CLO was detected at low concentrations (0.03 and 0.11 ng/g) even in samples collected from in nature conservation area. The results suggest that neonicotinoids may move off-target by wind or by erosion of contaminated soil.

Low levels of insecticide residues may remain in soils even upon years after the contamination event, and these residue levels tend to plateau to a mean concentration in the ng/g range. Estimated half-lives for TMX in 18 European soils ranged between 7.1 and 92.3 days [23]. The rate of dissipation is affected by microbial activity, resulting in CLO as a metabolite, being more persistent than and as toxic as the parent compound TMX. In contrast to slow dissipation of CLO in aerated soils, degradation is fast under anoxic soil conditions (e.g., flooded soils). Neonicotinoid insecticides proved to be persistent in the tropical soils with estimated half-lives above 150 days [24]. A study on the adsorption of CLO and TMX in three different agricultural soils in the state of Mississippi [25] reported strong positive correlations between the adsorption distribution coefficients and organic carbon content of the soils. In contrast, desorption was irreversible. The degradation of CLO and TMX in the soils was found to be slow with half-lives ranging from 100 to 280 days and 80 to 170 days for CLO and TMX, respectively. The degradation rates negatively correlated with the organic carbon content, but the moisture content in the soils had a positive effect on the degradation. Groundwater Ubiquity Scores calculated from the adsorption distribution coefficient, organic content and half-live suggests that CLO and TMX have moderate to high potential to permeate to groundwater.

According to a simple model calculation [26], the estimated half-life based on 8 year of crop history ranged between 0.25 and 2.12 years. The estimated mean half-life, based on measured neonicotinoid residues in the fields, was 0.4 year (ranging between 0.27 and 0.6 year). If CLO and TMX were used annually as a seed treatment in a typical crop rotation, residues would plateau rather than continue to accumulate after 3 or 4 year reaching a mean concentration of less than 6 ng/g in agricultural soils in southwestern Ontario.

The persistence of these AIs was confirmed by a recent study, in which neonicotinoid residues in nectar were quantified from winter-sown oilseed rape in western France collected within five years under the EU moratorium. All three restricted neonicotinoids (IMI, TMX, CLO) were detected. IMI showed no clear declining trend, but a strong inter- and intra-annual variation and maximum concentrations exceeded the reported concentrations in treated crops. Residue levels depended on soil type and increased with rainfall. Soil residues of IMI diffuse on a large scale in the environment and substantially contaminate major mass-flowering crops [27].

The environmental fate of neonicotinoids and their mobility in soils depends on soil properties as well [28]. We have studied the mobility of TMX and CLO along a soil column analyzing the AIs in the eluates (soil chromatography). High organic matter content retained even the more mobile ingredient (TMX) after the first period, whereas high clay content resulted in long release of CLO and TMX. Sandy soils with low organic content showed low retention capability; therefore, ingredients can easily leach and appear in surface and ground water.

Transport of neonicotinoids results in translocation of AIs via soil to plants emerged from non-coated seeds. Cross-contamination was detected by ingredient content of the guttation liquid (see below) and the rates depend on soil type as well [19]. When solution containing CLO and TMX were sprayed to the soil surface prior to emergence of non-coated seeds, ingredients were detected in the guttation liquid of non-coated plants grown in various soil types. CLO and TMX appeared in the guttation liquid immediately for plants emerged from non-coated seeds in sandy soil and somewhat delayed in clay. Loam soil retained the compounds for a longer period of time and they were detected 3 weeks after plantation of maize seeds.

It has been reported recently [29] that plants enhance vertical mobility of TMX from coated maize seeds in soil columns. Fine-particle soils transported over two orders of magnitude more TMX than coarse-textured soils (e.g., 29.9 μg vs. 0.17 μg, respectively), which was attributed to elevated evapotranspiration rates in the sandy soil driving a higher net retention of the pesticide and to structural flow occurring in fine-textured soil. Maize growth may drive preferential transport of TMX from coated seeds and may enhance neonicotinoid leaching, as the preferential transport along roots and root channels appears to have exceeded the ability of the plants to systematically uptake and retain TMX. This effect may become even more pronounced during short, high-intensity rainfall events. Even in the absence of preferential flow, TMX may become leached from soils over time.

## 4. Excretion via Guttation

Neonicotinoids, being systemic AIs, readily translocate into the xylem fluid, spread in the entire plant and appear in the guttation fluid as well. Guttation is a common phenomenon influenced by several factors. Exact levels of systemic pesticide AIs in guttation droplets vary by plant species, developmental stages and meteorological conditions as well [30]. Translocation of neonicotinoid insecticides from coated maize seeds to seedling guttation drops was discovered by Girolami and coworkers [31]. Although the excretion of neonicotinoids resulted in general rather variable concentrations in the guttation drops of maize, presumably due to environmental factors, high levels were observed during the first 3 weeks upon emergence. Neonicotinoid concentrations were measured up to 100 mg/L for TMX and CLO, and up to 200 mg/L for IMI [31]. The mean value for IMI was 47 ± 9.9 mg/L, the residue of CLO was 23.3 ± 4.2 mg/L and TMX was found in the guttation fluid at the level of 11.9 ± 3.3 mg/L.

Tapparo et al. [32] reported a decline of IMI concentrations in the guttation fluid of maize plants grown in the greenhouse, from 80.1 mg/L after the first day to 17.3 mg/L after 8–10 days, but the concentrations increased again to 60.1 mg/L during the next 10 days. During the first 6 days after emergence, IMI concentrations in the guttation drops collected at the top of the leaves ranged between 103 and 346 mg/L, while at the crown 8.2–120 mg/L were determined. Similar patterns were also seen for CLO (7.3–102 mg/L) and TMX (2.9–40.8 mg/L). TMX concentrations in the guttation fluid increased with decreasing soil moisture content, from 14 to 155 mg/L for plants grown under wet conditions to 34–1154 mg/L under dry conditions [32].

Our investigations [19,33] confirmed the levels reported earlier [31,32,34], but excretion of the guttation fluid was found not be limited to the first 3 weeks after germination. We have detected CLO even on the 45th day after emergence in the guttation fluid collected from plants emerged from coated seeds and grown in the field (see Figure 2). Fluctuation of the measured levels can be explained by the gentle air-flow in the field that results in occasionally intensified evaporation of drops and concentration of the guttation fluid at the leaf tips. Similarly, evaporation is influenced by relative humidity as well, and the extreme values seemed to correlate with both parameters. Thus, exact levels are affected not only by humidity, temperature, growth stage, etc., but also by the local air-flow (breeze) influences the concentration of the ingredients. To compensate for this process, drops were collected always in the same period of the day (between 7 and 8 A.M.).

In the laboratory experiments or if the plants were more protected from local air movement, smoother curves were obtained (see Figure 3) but the trends of curves show a decline in all cases. As the levels in the guttation drops are dependent on numerous factors, we used them in our experiments in parallel seedlings to characterize the differences in the ingredient uptake by plants.

Neonicotinoids were eliminated through the guttation fluid, as droplets on the leaf tip and also appeared in the middle of the leaves (crown cup). In accordance with a previous report [32], levels in the crown cup were significantly lower than in drops collected at the leaf tips (see Figure 3 and Figure 4). These parallel experiments were performed outdoor in pots. We have also observed an increasing trend after a decline at leaf edges (Figure 4). Comparison of CLO and TMX levels (Figure 3) showed higher concentrations for CLO than for TMX.

Translocation of residues in plants emerged from non-coated seeds has also been observed for plants that emerged from non-coated maize seeds but in close proximity with coated maize seeds [19]. High levels were determined for coated seeds, but cross-contamination via soil resulted in peak levels of 53.1 mg/L and 122.8 mg/L of CLO and TMX, respectively, for plants emerged from non-coated seeds nearby. After two weeks, the levels were similar for both types of plants. Overall amounts of CLO taken up by non-coated plants and detected in the guttation liquids were as high as 45.7% compared to that of plants emerging from coated seeds. Cross-contamination between coated maize seeds and two weeds (red poppy and creeping thistle) grown in close proximity to coated seeds has also been studied [33]. Translocation via soil was observed for both weeds, but the levels were substantially lower in the guttation liquid of the weeds than in that of maize plants that emerged from coated seeds.

Maximum concentrations of TMX were around 150 and 21 mg/L, while similar data for CLO were around 70 and 21 mg/L for maize and creeping thistle, respectively. Significant differences were found between the two weeds depending on the excreted volume of the guttation liquid. As red poppy guttate more intensively than creeping thistle, therefore the levels were lower in the former, reaching only 0.74 and 0.63 mg/L for CLO and TMX, respectively.

CLO was also detected in the guttation fluid of plants growing on a field next to a plot planted with coated maize seeds (1.25 mg/seed; Poncho) [35]. Samples collected at the field margin 1 or 24 h after sowing of the coated seeds were show to become contaminated via dust. All guttation fluid samples contained CLO at low levels up to 30 µg/L, with an overall average of 15.9 µg/L.

Contamination of non-target plants growing in the crop field margins [36] were also detected in 52% of the foliage samples collected. The average total concentration of neonicotinoid residues was found to be 10 ± 22 ng/g in wild plant samples grown in oilseed rape field margins. However, the median values of total neonicotinoids were higher in oilseed rape foliage (3.30 ng/g, range: 1.4–11 ng/g) than in wild plants (0.10 ng/g) due to highly variable quantities of residues, ranging between non-detectable levels to >106 ng/g for TMX. The results suggest that neonicotinoid seed-dressings lead to widespread contamination of the foliage of field margin plants with mixtures of neonicotinoid residues.

Only a part of the neonicotinoids taken up by maize seedlings can be measured in the guttation drops. The rate of the amount of ingredients excreted compared to the amount used in coating ranged typically between 1–2%. CLO, as a decomposition product of TMX, was detected in almost all guttation fluids collected from plants that emerged from TMX coated seeds. The ratio of the two insecticide AIs increased initially, but after two or three weeks the ratio became constant (about 30% for CLO/(CLO+TMX)) independently of the actual levels.

## 5. The Presence of Neonicotinoids in Surface Waters

Neonicotinoids have lately become ubiquitous contaminants in surface waters with high detection frequencies, but the average concentrations generally fall in the low ng/L range. Five compounds were selected in the EU as pollutants that may pose a significant risk at EU level to or via the aquatic environment [37]. They were also included into the second watch list of substances for EU-wide monitoring in the field of water policy [38]. Neonicotinoid levels in surface water were reviewed earlier by Morrissey and coworkers [39]. A number of monitoring studies reported the occurrence and levels of neonicotinoids in surface water (Figure 5) [22,39,40]. Elevated detection frequencies (98.7%–100%) and maximum concentrations were reported from high maize and soybean producing regions, and the transport of the residues was found to be driven also by precipitation.

In a nationwide study [48], at least one neonicotinoid AI was detected in 53% of the water samples collected from streams across the US. IMI was detected most frequently (37%), followed by CLO (24%), TMX (21%), dinotefuran (13%) and ACE (3%), whereas TCL was not detected. IMI occurrence was more related to urban uses, while CLO and TMX concentrations were significantly related to the amount of cultivated crops.

The occurrence of neonicotinoids was investigated along the St. Lawrence River, Canada in 2017 [49]. Among the targeted neonicotinoids, TMX and CLO showed the highest concentrations, respectively at <1–42 and <1–70 ng/L and average concentrations were of ∼4 ng/L each. IMI was also detected, but at lower concentrations (1.2–11 ng/L). Overall, 31% of the samples were found to surpass the guideline value of 8.3 ng/L for the sum of six priority neonicotinoids, proposed as chronic exposure criterion for aquatic wildlife in Quebec. Exceedances were more often observed in tributaries (67%) compared to the St. Lawrence River (22%). Similarly, detections were less frequent in the St. Lawrence River itself (55%), than in its surveyed tributaries (86%), and the average concentration for the sum of 6 priority neonicotinoids was of 3.7 ng/L within the St. Lawrence, while 23 ng/L in the tributaries.

Significantly higher detection frequencies were observed southern Ontario, Canada [50]. Of the five neonicotinoids studied, IMI, CLO and TMX exhibited detection rates above 90% at over half the sites sampled over a three-year period (2012–2014). At two sites, freshwater guideline value (acute exposure) for IMI (230 ng/L) was exceeded in roughly 75% of the samples collected. IMI concentration ranged from low ng/L to 10.4 µg/L. ACE and TCL were the less ubiquitous compounds, and concentrations rarely exceed the criterion value of 230 ng/L. CLO was found as ubiquitous pollutant with detection rates over 80% of samples at 10 of the 15 sites in the study and similar rates were obtained for TMX as well.

According to a study on Red River, Canada [51] neonicotinoids had relatively constant concentrations, suggesting a more widespread agricultural use both in the US and Canada. The maximum values for the three neonicotinoid insecticides TMX, CLO and IMI ranged 14.1–31.7 ng/L with mean concentrations over the two-year study <8 ng/L. Authors concluded, that levels of neonicotinoids posed minimal risks to aquatic organisms.

Neonicotinoids can be transported also into sources of drinking water in agricultural regions with heavy use of this class of insecticides. For instance, the most frequently used three neonicotinoids (CLO, IMI and TMX) were detected at concentrations as high as 57 ng/L in water collected from the drinking water treatment system for the University of Iowa in the US [52]. Similarly, these AIs appeared in drinking water in southern Ontario, Canada [53], however, the frequency of detection was much lower in treated drinking water than in raw drinking water. Estimated concentrations were in the low ng/L range in treated drinking water, but in one raw drinking water sample a mean concentration of 0.28 mg/L was determined.

Monitoring in the central Yangtze River, China in 2015 also resulted in high detection frequencies (64–100%) [54], and among six neonicotinoids, ACE, IMI and TMX were the most frequently detected ones. IMI had the highest median concentration of 4.37 ng/L, followed by ACE (2.50 ng/L), TMX (1.10 ng/L), nitenpyram (0.34 ng/L), CLO (0.10 ng/L) and TCL (0.02 ng/L), whereas the maximum concentrations reached the level of 236 ng/L and 44.4 ng/L for TMX and IMI, respectively.

Monitoring data of water organic pollutants identified by EU guidelines [37,38] were reviewed recently by Sousa et al. [55]. IMI was determined at up to 4560 ng/L in Australia and up to 1660 ng/L in Rio Grande do Sul, Brazil. TCL was also measured in Australia at up to 1370 ng/L. Neonicotinoid concentrations in surface waters in Europe seem to be lower. IMI level was found as high as 656 ng/L in Spain. TMX, IMI and TCL were found in the Pinios River basin, Greece at maximum concentrations of 330 ng/L, 306 ng/L and 120 ng/L, respectively. IMI was found among the most frequently detected analytes in a monitoring survey of the Watch List contaminants in the Ave and the Sousa Rivers, Portugal [56]. TMX was detected in both rivers, while CLO was found only in the Ave and TCL in the Sousa. IMI and TMX were found in the Ave during all seasons, with the highest concentrations in the summer at up to 480 ng/L and 215 ng/L, respectively. These two neonicotinoids were detected in the Sousa in the summer and autumn sampling campaigns, although at lower concentrations: IMI at up to 208 ng/L (summer) and 213 ng/L (autumn), and TMX at up to 4.7 ng/L (autumn) and 17.8 ng/L (summer). CLO was only quantified in two sampling campaigns (winter and summer) in the Ave, at up to 51.7 ng/L.

Our earlier results [57] indicated lower levels of neonicotinoids (TMX 4–30 ng/L, CLO 17–40 ng/L) as diffuse contaminants in Hungarian surface waters, higher levels (10–41 µg/L) were found only sporadically for CLO in ponds near crops emerged from treated maize and sunflower seeds. After restriction of seed coating by three neonicotinoids (IMI, TMX and CLO) in 2013, we monitored the Danube River in the late spring and early summer 2017. Higher concentrations are expected in the summer, which can be a consequence of the leaching effects promoted by the agriculture activities and by precipitation. Contamination rates were 36% in this period and concentrations (See Table 2) ranged 4.65–16.83 ng/L and 3.54–11.43 ng/L for TMX and CLO, respectively, in the Danube River at Budapest, Hungary. In general, these concentrations well correlate with other neonicotinoid occurrence data in large rivers.

The presence of certain neonicotinoids was found to influence the loss of linear alkylbenzensulfonates (LASs), and loss rates were found to depend also on the aqueous matrix (distilled water or surface water collected from Danube River). LASs are widely used surfactants, e.g., as formulating agents of Mospilan [58]. Loss of LAS homologues (alone and in the presence of Mospilan 20 SG, containing ACE and LASs as well) was determined in two aqueous matrices. Decomposition of LASs was more rapid in surface water from Danube River than in distilled water. Decomposition rates depended also on the presence of ACE (as Mospilan 20 SG formulation) in surface water from Danube River. DT_50_ values were found to be 58.7 ± 2.0 h and 495 ± 32 h for LAS alone and in formulation in surface water, respectively. Similar tendencies were obtained in distilled water for LASs alone and in Mospilan 20 SG, but some differences were observed between the corresponding DT_50_ values (215 ± 18 h and 519 ± 32 h).

The presence of other neonicotinoid AIs together with LASs in surface water from Danube River either had no effect on the loss of LASs (TMX, CLO) or affected the loss rates negatively (ACE, TCL, IMI). Averages of DT_50_ values were found to be 77.4 ± 13.5 h, 75.5 ± 11.0 h and 79.4 ± 12.3 h for LASs alone and in the presence of TMX or CLO, respectively. In the presence of ACE 107 ± 4 h were calculated from the curves, whereas nearly doubled values, 146 ± 5 and 139 ± 6 h were obtained for DT_50_ of LASs in the presence of TCL and IMI, respectively.

## 6. Ecotoxicological Testing of Neonicotinoid Active Ingredients and Their Formulations

### 6.1. Toxicity of Neonicotinoid Active Ingredients and Formulations on Daphnia magna

PPPs applied in chemical plant protection contain various co-formulants, beside their AIs, which co-formulants have been regarded as inactive/inert components, however have been proven in numerous studies to cause disadvantageous side effects or affect the toxicity of the AIs [59,60,61,62]. Moreover, as discussed above, globally used neonicotinoids and components of their formulated PPPs, due to their physical-chemical properties (e.g., water solubility) or improper applications, can reach surface waters and have adverse effects on the aquatic ecosystems and non-target organisms [63,64,65].

In our previous studies individual and combined toxicity of several neonicotinoid AIs and additives used in neonicotinoid PPPs were investigated on the widely used aquatic test organism *Daphnia magna*, a proper indicator of the detrimental effects of environmental contaminants on aquatic habitats due to its outstanding sensitivity to changes in water quality. Our acute tests performed according to OECD standards (OECD Test No. 202: *Daphnia* sp. acute immobilization test) [66] demonstrated the unexpected toxicity of assumedly inert formulating agents of PPPs. Significant differences were found in the toxicity of the neonicotinoid AIs and their formulations tested [58,67]. The individual toxicity of AIs TCL and TMX were 2.4 and 1.9 times higher than their formulations investigated, respectively, while the toxicity of CLO and ACE was significantly higher in the presence of the additives used in their PPPs. The neonicotinoid formulation containing LASs as additives was found to be 46.5 times more toxic than its AI CLO itself on *D. magna* [67]. As demonstrated by our results, formulating agents can enhance the toxic effects of CLO and ACE or reduce the toxicity of TCL and TMX in their formulated PPPs [67]. Similarly, as in our results, possible effects of additives (e.g., surfactants) on the toxic effect of AIs were detected in other studies on various aquatic indicator species as well [61,68,69]. Acute testing was completed with the determination of combined effects of ACE and LASs used as AIs and the formulating agent in Mospilan 20 SG. The combined toxicity of the two components was tested in the form of the investigated insecticide formulation and in the form of the mixture of their pure forms according to the composition of Mospilan 20 SG. The individual toxicity of LASs was significantly higher on *D. magna* compared to the individual toxicity of ACE, and the toxic effect of the formulation was found to be 1.3 and 19.6 times higher than explained by its AI and LAS content, respectively, indicating synergistic toxicity. The strongest synergy between ACE and LASs was detected, when the pure forms of the investigated components were used in combination at concentrations equivalent to those in Mospilan 20 SG. The higher toxicity of surfactants was also observed by several studies [61,70,71] on *D. magna* as well [72,73,74], but the acute toxicity of LASs was highly affected by the length of alkyl chain and molecular weight of LASs on *D. magna* [75]. Synergistic toxicity was observed between ACE and other alkylphenol ethoxylates as well [73].

In addition to the acute immobilization tests, the activity of the detoxifying glutathione S-transferase (GST) enzyme was also investigated according to the method described by Habig et al. using microtiter plates as well [76] in *D. magna* juveniles exposed to Mospilan 20 SG and its components. During enzyme activity assays the individual and combined toxicity of ACE and LASs was investigated in the form of their pure mixture and the formulated insecticide similarly to the acute testing at the concentrations equivalent to one third of the calculated EC_50_ values after 48 h exposition. The activity of GST at the applied ACE concentration (65 mg/L) was significantly lower compared to the control unit (*p* < 0.01), while in the immobilization tests the toxic effects of the AI was not detected as ACE at the concentration of 200 mg/L caused only 10% immobility in the exposed juveniles. Decreased activity of GST was detected in test groups exposed to the combinations of the two investigated components in both forms, but only the pure mixture caused a significant reduction in enzyme activity (*p* < 0.05) (Figure 6). The combined toxicity of ACE and LASs resulted in lower enzyme activity compared to the control and groups exposed to LASs (Mospilan: *p* < 0.05; ACE-LAS: *p* < 0.01), but it was not significantly different from the juveniles treated with ACE alone, and significant differences were not observed between the tested combinations (*p* > 0.05). A slight increase in detoxifying enzyme activity was observed in the group exposed to LASs compared to the control. In contrast to the immobilization tests, the high toxicity of LASs was not detected on GST activity at the investigated concentration of 3.8 mg/L. Compared to the observed individual effects of LASs, lower LAS concentrations (0.93 mg/L) resulted in inhibited enzyme activity in the presence of ACE at concentration (7.4 mg/L) equivalent to those in Mospilan 20 SG (Figure 6).

On the basis of our results the toxicity of ACE was higher in GST enzyme activity assays compared to the acute immobilization tests and caused the reduction or inhibition of GST activity. The presence of ACE individually or in combination with LASs resulted in decreased enzyme activity in the exposed groups. The inhibition of GST activity was proven on other test organisms (*Eisenia andrei* and *Apis mellifera*) at higher concentrations [77,78] and on *D. magna* as well after the exposition of other neonicotinoid and pesticide AIs [79,80]. The reduction of GST activity is probably caused by the overproduction of reactive oxygen species (ROSs) as a result of xenobiotic-induced oxidative stress, while the released ROS significantly delay or inhibit the oxidation of substrate [81].

Our results indicate that co-occurrence of the investigated AIs and surfactant has significant effects on their combined toxicity, while additives used for the formulation of pesticide products may change the effect of the AIs on non-target species. Consequently, formulating agents applied in agrochemicals cannot be considered as unequivocally inactive ingredients from toxicological and ecotoxicological aspects.

### 6.2. Neuronal Targets of Neonicotinoids in the Snail Nervous System

Considering the abundant toxicological data of neonicotinoids on aquatic insect and crustacean taxa [39,82,83,84], non-arthropod members of the aquatic ecosystem including molluscs are often neglected in field surveys or laboratory studies. One possible reason for it is, that molluscs (both aquatic and terrestrial) are generally rather resistant to numerous insecticides (organophosphates, carbamates or pyrethroids) and their residues in water [85,86,87]. The sensitivity differences may be due to different toxin–target interactions and heterogeneity of the toxin-induced changes in metabolism or gene expression [85,86,88]. Recent results also suggest contribution of the cellular defense system (the multixenobiotic resistance mechanism, MXR) in the organism’s resistance to neonicotinoid insecticides [89].

The morphological and physiological features of the snail nervous system have numerous advantages for testing potential neurotoxins, including neonicotinoid insecticides. The easily visualized giant neurons allow to perform repeated experiments on virtually identical neurons in the central nervous system (CNS) as well as to characterize the toxin target interactions on the cellular/membrane level by single cell recording techniques [90,91,92].

Neonicotinoids, the third-generation nicotinic insecticides impair the cholinergic neurotransmission in the central nervous system [93,94], and recently it is also confirmed that they similarly target the nicotinergic acetylcholine receptors (nAChRs) in the CNS of the pond snail, *Lymnaea stagnalis* [64]. To further characterize the neuronal effects of neonicotinoids in snails, we performed electrophysiological experiments on identified central neurons of the terrestrial snail *Helix pomatia*, a well-established model to study toxin-target interactions on membrane level [95,96,97]. While recording single cell activity of the identified RPas neurons by conventional electrophysiological methods [98], acetylcholine (ACh) or its analogue carbachol (Carb) were injected locally to membrane and the insecticides in the form of their commercially available products (Mospilan 20 SG, Kohinor and Actara, containing neonicotinoid AIs ACE, IMI and TMX, respectively) were applied extracellularly in the bathing medium.

Local application of 1 mM ACh from a micropipette strongly excited all members of the RPas cluster neurons resulting increased frequency of spontaneous firing, often followed by short inhibition (Figure 7A1). The ACh analogue carbachol (100 μL, 1 mM) in the bath similarly increased the frequency of the intracellular activity of the neuron (Figure 7A2). After a similar application of the neonicotinoid product Mospilan (100 µL, 1 mg/mL, equal with 0.9 mM of its AI, ACE), no intracellular activity change was recorded (Figure 7A3), but resulted a strong inhibition of the ACh-evoked inward current (Figure 7B1,B2). Local application of 1 mg/L neonicotinoid products Actara and Kohinor (containing TMX and IMI as AIs, respectively) did not evoke any direct membrane effects (alteration of the spontaneous activity) either (not demonstrated).

After these insecticides (Actara and Kohinor) were applied extracellularly in the bath, they similarly inhibited the ACh-evoked inward currents in a dose-dependent way (Figure 8). While comparing their efficiency, Mospilan and Actara (containing ACE and TMX, respectively) were found to have about the same inhibitory potential (ACE: IC_50_ = 2.03 mg/L or 9.1 μM, *n* = 12; TMX: IC_50_ = 3.41 mg/L or 11.7 μM, *n* = 14), and Kohinor (IMI) was slightly less effective to block the ACh responses (IMI: IC_50_ = 8.87 mg/L or 34.7 μM, *n* = 11).

Detailed dose-response studies also revealed an additional effect, showing that in lower (3–20 µM) concentrations of their AIs all the neonicotinoid insecticides slightly increased the amplitudes of the ACh-evoked currents were a slightly increased (Figure 9A–C, left) while applying higher (30–150 µm) concentrations in the bath all the tested insecticides decreased the amplitudes of the inward currents (Figure 9A–C, right).

Local application of the ACh analogue carbachol (1 mM) from a micropipette also evoked an inward current, but the bath applied neonicotinoids (at any concentrations) always inhibited, never enhanced the ACh-evoked membrane responses. (Figure 10A–C).

In insects, neonicotinoids are considered as nAChR agonists which initially mimic the effect of ACh, but overstimulating the receptors also prevent their response to the naturally released neurotransmitter [93,99]. During our experiments on *Helix* neurons however, we never observed ACh agonist (excitatory) effects when neonicotinoids were applied locally. These results well agree with our previous data on *Lymnaea* neurons [64], that neonicotinoid insecticides lack of cholinergic activity (ACh-like agonist effects) on snail neurons. The inhibitory effects of neonicotinoid insecticides however, were clearly demonstrated the on cholinergic responses, either evoked synaptically (in *Lymnaea*), or by local ACh application of (both in *Helix* and *Lymnaea*). These results suggest a special feature of the snail nACRs confirming the pharmacological difference between snail and insect nAChRs.

Dose-response analysis demonstrated about the same potency of all the three neonicotinoid products we tested (EC_50_ values ranging between 9.1 μM (ACE) and 34.7 μM (IMI) corresponding with product concentrations of 10 mg/L (Mospilan) up to 43.5 mg/L (Kohinor) respectively). These values of neonicotinoid PPPs are in the same range of concentrations (10–100 mg/L) we applied previously to inhibit ACh responses of *Lymnaea* neurons [64]. Enhancement of ACh-evoked inward currents in the presence of neonicotinoids (Mospilan, Actara, Kohinor), however, seem to be a new feature of the neonicotinoid effects, some additional mechanism of toxin-target interactions, not reported previously. Enhancement of Ach-evoked inward currents in the presence of neonicotinoids (Mospilan, Actara, Kohinor), however, seem to be a new feature of the neonicotinoid effects, suggesting some additional mechanism of toxin-target interactions, not reported previously. Increased amplitudes the ACh-evoked membrane currents can be resulted by inhibiting the acetylcholine esterase enzyme (AChE), which contributes in breakdown of the synaptically released ACh in the synaptic cleft. Accordingly, inhibition of the enzyme may increase the ACh effects on snail neurons. This suggestion seems to be confirmed by the lack of enhancing effects when the inward current is evoked by carbachol application, which molecule is resistant to AChE, i.e., not metabolized by this enzyme. Although we cannot provide direct evidence now, literature data confirm our suggestions. Altered AChE activity was reported in the tissues of a number of invertebrates treated by neonicotinoids including insects [100,101,102], earthworm [103], mussel [85] and snail [104]. We cannot exclude, therefore, that neonicotinoids interact with both the nAChR and the AChE enzyme, which share a common substrate, the ACh molecule. Simultaneous interaction of pesticides with the AChE enzyme and the nAChRs were already demonstrated in house fly brain by radioligand binding [105], on rat 4b2 nAChRs [106] and on electric eel nAChRs [107].

In summary, we demonstrated the antagonist effects of neonicotinoid pesticide products on the snail nAChRs, but unlike in insects, no agonist (ACh-like) effects were confirmed. In addition, enhancement of the excitatory responses (inward currents) by lower concentration of neonicotinoids confined to the ACh responses, while similar carbachol-evoked currents were only inhibited by neonicotinoids. The above experimental data suggest heterogeneous molecular targets (nAChR and AChE) of neonicotinoid insecticides (ACE, IMI, TMX) in the snail CNS.

### 6.3. Toxicity Tests of Neonicotinoids on other Indicator Organisms

In addition to the above aquatic exotoxicity tests, neonicotinoid PPPs Calypso 480 SC and Mospilan 20 SG and their AIs TCL and ACE, respectively, were also evaluated in a standardized limit test (100 mg/L) according to the corresponding ISO standard 11269-2 [108] to determinate the effects of the investigated substances on the emergence and early growth of higher terrestrial plants as garden cress (*Lepidium sativum*), white mustard (*Sinapis alba*), common wheat (*Triticum aestivum*). No significant differences were observed on root length of *S. alba* among treatments, however the investigated AIs ACE and TCL, moreover the TCL-based formulation induced plant growth by 23.1%, 39.2% and 23.8%, respectively. Inhibited root growth by 24.0% and plant growth by 18.4% was observed in *T. aestivum* exposed to Mospilan 20 SG. Exposition to Calypso 480 SC also resulted in the inhibition of plant growth (15.0%) in common wheat. Differences on the root lengths were not observed in *L. sativum*, but 22.0% inhibition of plant growth was detected in the groups exposed to Calypso 480 SC compared to its AI TCL.

## 7. Conclusions

In spite of their outstanding specificity towards insects, neonicotinoids have made a substantial negative impact on the environment and show environmentally unfavorable features: surface water contamination potential, persistence in soil and in plants (e.g., in the guttation liquid), translocation in soil from seed coating materials to affected plants (crop or weeds), as well as toxicity to non-target organisms (not only bees as a major insect family at risk, but also aquatic crustaceans (*D. magna*) and mollusks (*L. stagnalis* and *H. pomatia*) as indicated in the present study). In addition, elevated exposures to neonicotinoids due to their increasing dosages in seed coating and high rate of use in crops further enhance the risks they pose.

Bans or restrictions in their application raise the question of suitable alternatives to their use, including biological, semiochemical and physical methods through altered agrotechnology e.g., more proper timing of applications by considering pest life-cycles and habitat characteristics, as well as repressing the rising application volumes that resulted from undue expansion in the use of seed coating. Nonetheless, negative consequences of those restrictions also occurred: the ban of neonicotinoids resulted in a boost of the use of remaining organophosphate insecticides (particularly chlorpyriphos) posing potentially more severe risks than those associated with the restricted neonicotinoids. In turn, this resulted in the acceleration of the legislative constraints intended against the replacement substances, manifested in the form of an EU-wide ban of chlorpyriphos and chlorpyriphos-methyl.

There seems to be a general consensus among members of the Task Force on Systemic Pesticides that alternative methods beyond a modified use pattern of agrochemicals is an essential part of sustainable agriculture, and such appropriate methods can ensure satisfactory levels of pest suppression. On the other hand, a more restrictive regulatory framework is required to put the integrated pest management principles into practice, and further development of applied agroecology will also broaden the range of new alternative pest control strategies.

There is no consensus, however, regarding alternative insecticide AIs to replace neonicotinoids; and increasing apparent resistance to pyrethroid insecticides together with the ban on neonicotinoid seed dressing and withdrawal of chlorpyriphos essentially impact pesticide use in the EU. No effective solution can be expected from insecticides acting by the same mode of action (MoA) as neonicotinoids (nicotinic acetylcholine receptor agonists e.g., sulfoximines, butenolids, spinosoids) as their side-effect pattern, particularly toxicity to bees, is expected to be similar. Possible alternatives to some extent could be novel insecticides with other MoA, such as anthranic diamides (ryanoids) (e.g., chlorantraniliprole, cyantraniliprole) or tetronic acid derivatives (e.g., spiridiclofen, spiromesifen, spirotetramat). Of these AIs, however, solely cyantraniliprole is registered not only as as a spray insecticide, but also to be applied in seed coating, and even this authorization applies only for cole crops.

## Figures and Tables

**Figure 1 ijerph-17-02006-f001:**
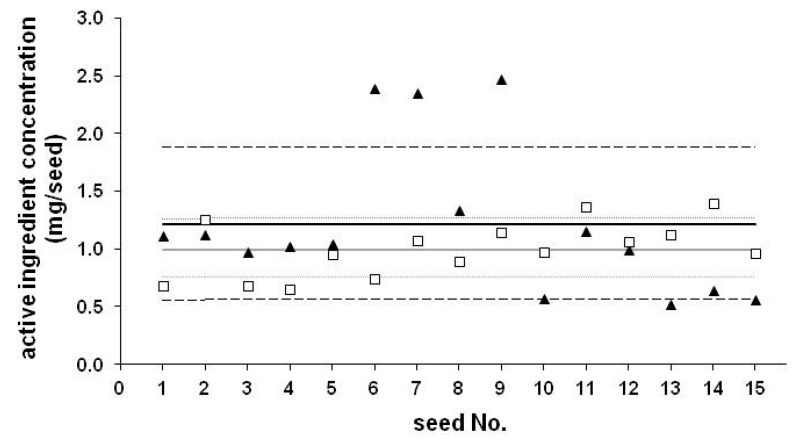
Concentration of clothianidin (CLO) on individual seeds (mg/seed), CS-3 (square) and CS-4 (triangle). Average concentration determined for 15 seeds (continuous lines) and the relative standard deviation (RSD) (dashed lines).

**Figure 2 ijerph-17-02006-f002:**
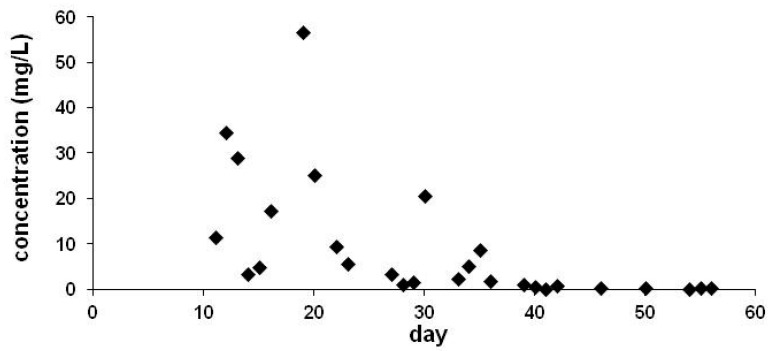
Concentration in the guttation liquid of maize plants emerged from coated seeds (0.997 mg CLO/seed) under field conditions.

**Figure 3 ijerph-17-02006-f003:**
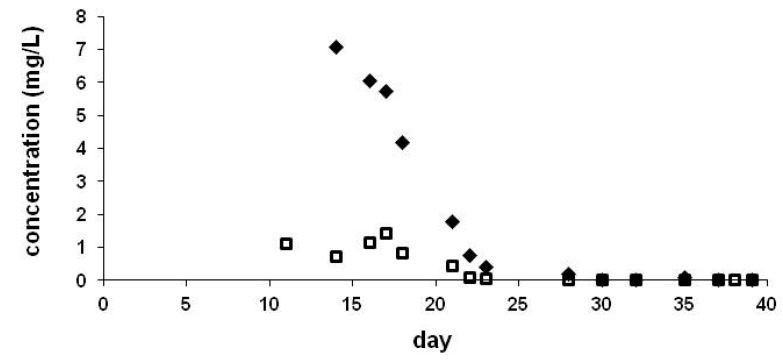
Concentration of CLO (♦) and thiamethoxam (TMX) (**□**) in the guttation liquid collected from crown cup. Maize seeds were coated by 0.605 mg/seed TMX or 1.217 mg/seed CLO, experiments were performed outdoors in pots.

**Figure 4 ijerph-17-02006-f004:**
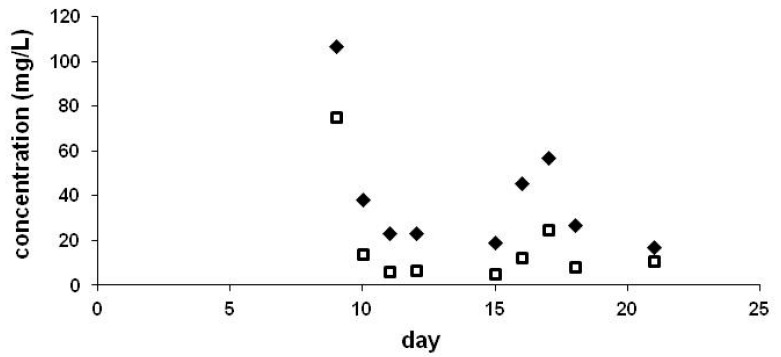
Concentration of CLO (♦) and TMX (□) in the guttation liquid collected at leaf edges of maize plants. Maize seeds were coated by 0.605 mg/seed TMX or 1.217 mg/seed CLO, experiments were performed outdoors in pots.

**Figure 5 ijerph-17-02006-f005:**
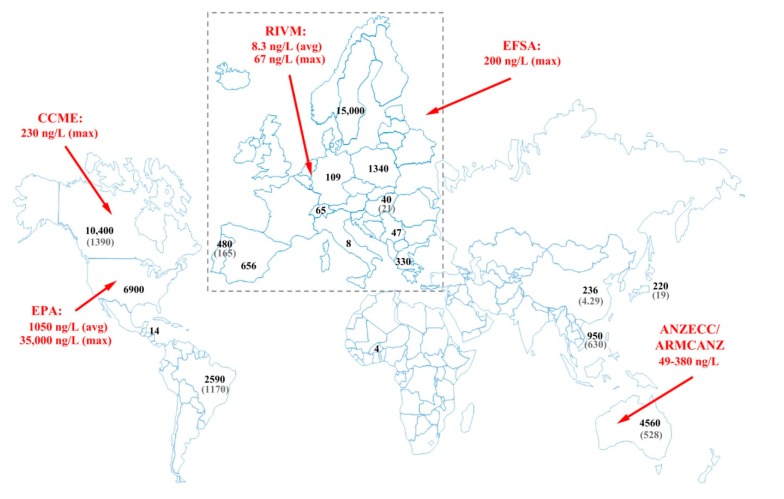
The maximum and the corresponding average concentrations in surface waters reported worldwide. Data are referred in this publication or in ref. [22,39,40]. The indicated quality reference values: European Food Safety Authority (EFSA) no observable effect concentration (NOEC) [41], Rijksinstituut voor Volksgezondheid en Milieu (RIVM) Nederland maximum permissible concentration and long term exposure [42,43], US-EPA aquatic life benchmark [44], Canadian Council of Ministers of the Environment (CCME), water quality guideline [45], Australia and New Zealand proposed aquatic ecosystem protection guideline values depending on species protection goals (80%–99%) [46] and a default guideline value proposed by third parties is under peer review process [47].

**Figure 6 ijerph-17-02006-f006:**
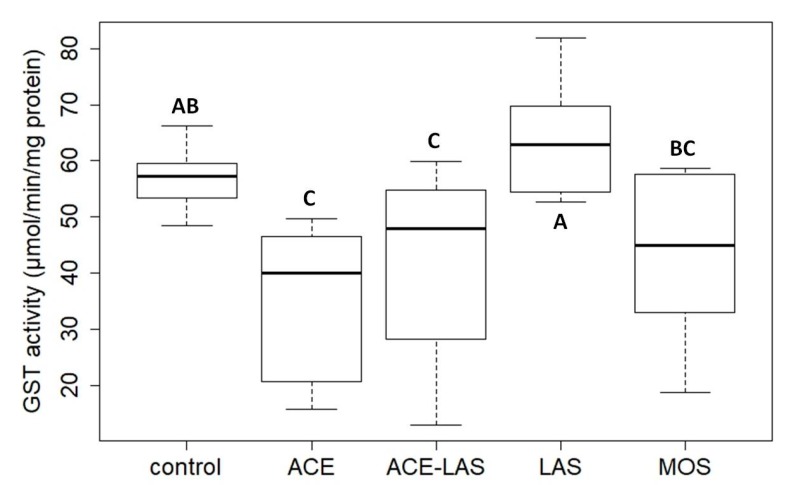
Activity of glutathione S-transferase (GST) enzyme in *D. magna* juveniles exposed to Mospilan 20 SG (MOS) and its components (active ingredient (acetamiprid (ACE)) and formulating agent (linear alkylbenzensulfonates (LASs))) individually and in combinations at concentrations equivalent to one third of the calculated 48 h EC_50_ values according to the results originated from immobilization tests. Combined toxicity of the investigated components was investigated in the form of their pure mixture (ACE-LAS) and formulated insecticide (MOS).

**Figure 7 ijerph-17-02006-f007:**
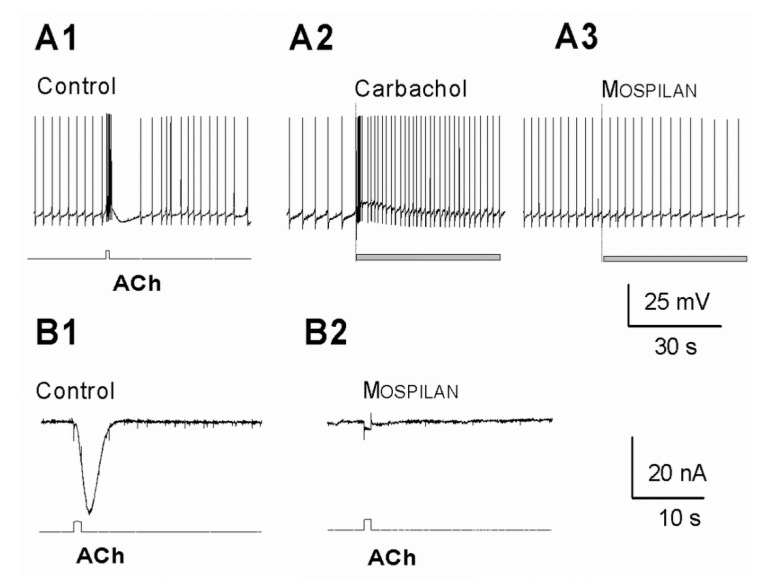
Intracellular responses of the identified *Helix pomatia* neurons. In normal saline (control) local application of 1 mM ACh evokes strong intracellular response (**A1**) and carbachol (100 μL, 1 mM) similarly increases the firing frequency (**A2**), while Mospilan (100 μL, 1 mg/mL) does not alter the spontaneous activity (A3). (**B**) Voltage clamp recording showing that the ACh-evoked inward current (**B1**) is virtually abolished in the presence of 0.1 mg/mL Mospilan when applied in the bath (**B2**).

**Figure 8 ijerph-17-02006-f008:**
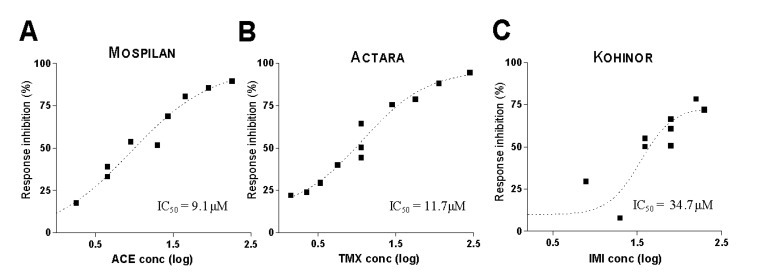
Dose-response analysis of neonicotinoid effects showing Mospilan (**A**), Actara (**B**) and Kohinor (**C**) all inhibiting the ACh-evoked inward currents. X axis: amplitude values in percentage of the control response. Y: Log concentrations of insecticides in molarities of their active components.

**Figure 9 ijerph-17-02006-f009:**
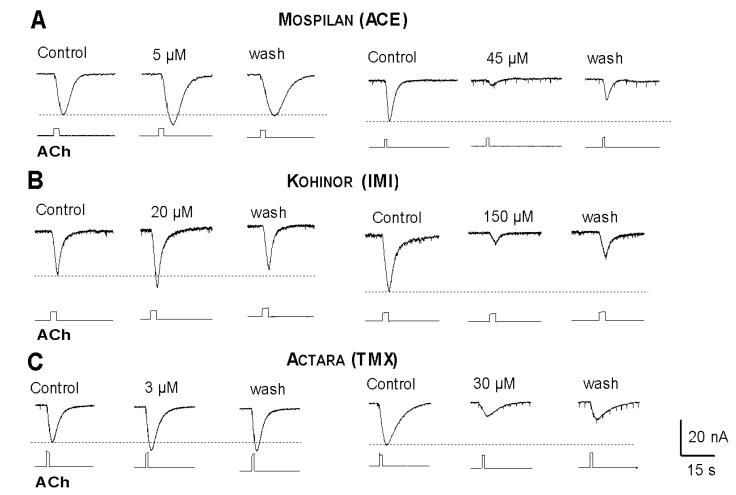
Dose dependent alterations of ACh-evoked inward currents in the presence of Mospilan (**A**), Kohinor (**B**) and Actara (**C**). In lower concentrations of their active ingredients the amplitudes are slightly increased (on the left) while at higher concentrations all neonicotinoids inhibit the ACh responses (on the right). Dotted lines mark the amplitudes of ACh currents in control (physiological saline without insecticides). Concentration values are expressed in molarities of the active components (e.g., ACE in Mospilan).

**Figure 10 ijerph-17-02006-f010:**
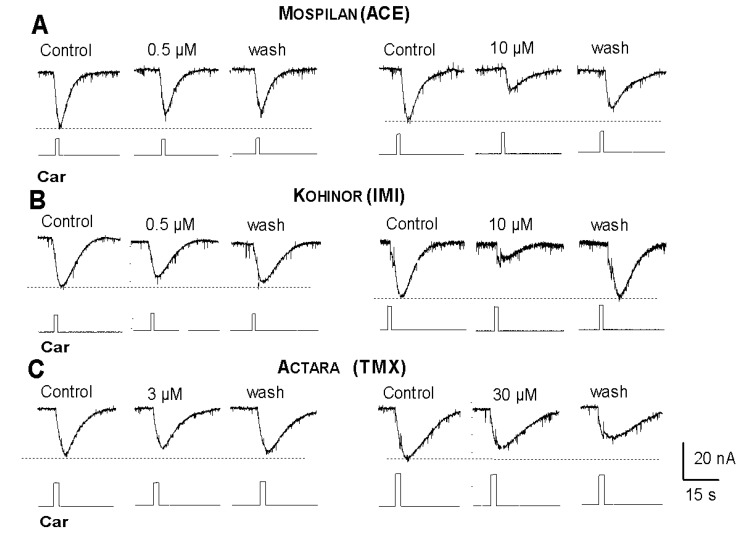
Inhibitory effects of neonicotinoids on ACh-evoked inward currents by Mospilan (**A**), Kohinor (**B**) and Actara (**C**). Concentration values are expressed in molarities of the active components of the individual insecticides.

**Table 1 ijerph-17-02006-t001:** The concentration of neonicotinoid active ingredients (AIs) in seed coating of maize seeds.

Code	AI	Concentration (mg/seed)	RSD (%)	Number of Seeds	Year	Recommended Dose (mg/seed)	Typical Doses (mg/seed)
CS-1	TMX	0.29	18.6	10	2007	0.125–1.25	1.26 or 0.63
CS-2	TMX	0.26	15.3	10	2007	0.125–1.25	1.26 or 0.63
CS-3	CLO	0.997	24.2	15	2009	0.25–1.25	1.25
HC-1	CLO	0.61	16.8	10	2014	0.25–1.25	1.25
HC-2	TMX	0.15	20.5	10	2014	0.125–1.25	1.26 or 0.63
HC-3	TCL	0.054	9.7	10	2015	1.00	1.00
CS-4	CLO	1.217	54.3	15	2013	0.25–1.25	1.25
CS-5	TMX	0.605	12.3	10	2013	0.125–1.25	1.26 or 0.63
CS-6	TCL	1.18	11.2	10	2015	1.00	1.00

**Table 2 ijerph-17-02006-t002:** Concentration of neonicotinoids (ng/L) determined in Danube River at Budapest, Hungary in 2017.

Code	Sampling Date	Concentration (ng/L)	
		TMX	CLO
V1265	May 4	<LOD	<LOD
V1266	May 11	<LOD	<LOD
V1267	May 18	<LOD	4.22
V1268	May 25	<LOD	<LOD
V1269	June 1	<LOD	<LOD
V1270	June 8	<LOD	<LOD
V1277	June 14	16.83	11.43
V1278	June 21	4.65	4.10
V1279	June 28	5.20	3.54
V1280	July 5	15.80	<LOD
V1281	July 11	<LOD	<LOD

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
