# Peer review of "Neonicotinoids: Spreading, Translocation and Aquatic Toxicity"

_ijerph, 2020, doi:10.3390/ijerph17062006_

Round 1

Reviewer 1 Report

I recommend this manuscript for publication in International Journal of Environmental Research and Public Health.

Author Response

Many thanks for the very positive opinion of reviewer.

Reviewer 2 Report

The review provides interesting information about the translocation and aquatic toxicity of necotinoid insecticides. The manuscript does provide some useful information that will be helpful in insecticide uses and provides additional experiences. Nevertheless, the manuscript is in need of revisions before it is acceptable for publication.

A few points:

Lines 30-31: Keywords should be in alphabetic order

Lines 50-63: In this paragraph, provide the main objective of this review

Line 111:…the use of less equipment…

Line 111: Place “.” after parenthesis

Line 194: …systemic…

Line 196: …systemic…

Line 267: …independently of the…

Line 387: …side effect…

Lines 572-573: …plants as garden cress…

Author Response

Point 1. Lines 30-31: Keywords should be in alphabetic order

Modified as requested and completed.

Point. 2. Lines 50-63: In this paragraph, provide the main objective of this review

Due to the modifications, the main objectives are given in lines 90-96, as requested. Added text in indicated by yellow in the revised manuscript.

Minor modifications have been done, as requested. New line numbers are indicated below.

Line 111:…the use of less equipment…  Line 143

Line 111: Place “.” after parenthesis  Line 139

Line 194: …systemic…  Line 227

Line 196: …systemic…  Line 229

Line 267: …independently of the…  Line 310

Line 387: …side effect…  Line 420

Lines 572-573: …plants as garden cress…  Line 606

Reviewer 3 Report

The authors have summarized the most recent information on ecotoxicological characteristics of neonicotinoids. The manuscript is well prepared, is readable and suitable for Int. J. Environ. Res. Public Health after making minor modifications.

Suggested modification: Given the negative impact of neonicotinoids on the environment and, at the same time, their high rate of use in plant protection, it would be advisable to give a short thought to suitable and environmentally safe neonicotinoid alternatives.

In the conclusion the authors should summarize (generalize) the most important facts on neonicotinoids and, as appropriate, propose any practical recommendations or directions for research.

Author Response

Point 1. Suggested modification: Given the negative impact of neonicotinoids on the environment and, at the same time, their high rate of use in plant protection, it would be advisable to give a short thought to suitable and environmentally safe neonicotinoid alternatives.

Response 1. A new paragraph has been inserted into the introduction about the possible alternatives including non-chemical and agroecological methods. As the new US EPA decision is also mentioned, altogether six new references were cited in this part.  The added text is indicated by yellow in the revised manuscript.

Point 2. In the conclusion the authors should summarize (generalize) the most important facts on neonicotinoids and, as appropriate, propose any practical recommendations or directions for research.

Response 2. We draw up conclusion in a new section, where we summarized the most important facts, discuss the situation after the ban in EU, and some recommendations are also formulated, as requested.  The added text is indicated by yellow in the revised manuscript.